# Associations of Glucometabolic Indices with Aortic Stiffness in Patients Undergoing Peritoneal Dialysis with and without Diabetes Mellitus

**DOI:** 10.3390/ijms242317094

**Published:** 2023-12-04

**Authors:** Chi-Chong Tang, Jen-Pi Tsai, Yi-Hsin Chen, Szu-Chun Hung, Yu-Li Lin, Bang-Gee Hsu

**Affiliations:** 1Division of Nephrology, Hualien Tzu Chi Hospital, Buddhist Tzu Chi Medical Foundation, Hualien 97002, Taiwan; dearedward1025@gmail.com; 2Institute of Medical Sciences, Tzu Chi University, Hualien 97004, Taiwan; 3School of Medicine, Tzu Chi University, Hualien 97004, Taiwan; tsaininimd1491@gmail.com (J.-P.T.); szuchun.hung@gmail.com (S.-C.H.); 4Division of Nephrology, Department of Internal Medicine, Dalin Tzu Chi Hospital, Buddhist Tzu Chi Medical Foundation, Chiayi 62247, Taiwan; 5Division of Nephrology, Department of Internal Medicine, Taichung Tzu Chi Hospital, Buddhist Tzu Chi Medical Foundation, Taichung 42743, Taiwan; nephp06@gmail.com; 6Division of Nephrology, Department of Internal Medicine, Taipei Tzu Chi Hospital, Buddhist Tzu Chi Medical Foundation, Taipei 23142, Taiwan

**Keywords:** impaired fasting glucose, insulin resistance, glucose load, aortic stiffness, non-diabetic, peritoneal dialysis

## Abstract

Disruptions in glucose metabolism are frequently observed among patients undergoing peritoneal dialysis (PD) who utilize glucose-containing dialysis solutions. We aimed to investigate the relationship between glucometabolic indices, including fasting glucose, insulin resistance, advanced glycation end products (AGEs), PD-related glucose load, and icodextrin usage, and aortic stiffness in PD patients with and without diabetic mellitus (DM). This study involved 172 PD patients (mean age 58.3 ± 13.5 years), consisting of 110 patients without DM and 62 patients with DM. Aortic stiffness was assessed using the carotid-femoral pulse wave velocity (cfPWV). Impaired fasting glucose was defined as a fasting glucose level ≥ 100 mg/dL. Homeostatic model assessment for insulin resistance (HOMA-IR) scores, serum AGEs, dialysate glucose load, and icodextrin usage were assessed. Patients with DM exhibited the highest cfPWV (9.9 ± 1.9 m/s), followed by those with impaired fasting glucose (9.1 ± 1.4 m/s), whereas patients with normal fasting glucose had the lowest cfPWV (8.3 ± 1.3 m/s), which demonstrated a significant trend. In non-DM patients, impaired fasting glucose (β = 0.52, 95% confidence interval [CI] = 0.01–1.03, *p* = 0.046), high HOMA-IR (β = 0.60, 95% CI = 0.12–1.08, *p* = 0.015), and a high PD glucose load (β = 0.58, 95% CI = 0.08–1.08, *p* = 0.023) were independently associated with increased cfPWV. In contrast, none of the glucometabolic factors contributed to differences in cfPWV in DM patients. In conclusion, among PD patients without DM, impaired fasting glucose, insulin resistance, and PD glucose load were closely associated with aortic stiffness.

## 1. Introduction

Peritoneal dialysis (PD), the primary home-based dialysis modality, constituting approximately 11% of renal replacement therapies for end-stage renal disease (ESRD) worldwide [1], offers comparable survival rates, improved quality of life, preserved residual renal function, and superior cost-effectiveness when compared to hemodialysis (HD) [2,3,4,5]. However, the widespread use of glucose-based dialysate solutions in PD leads to the inevitable absorption of glucose into the bloodstream. This raises concerns about increased glucometabolic burdens and disrupted glucose homeostasis, thereby elevating the risks of central obesity, insulin resistance, and new-onset hyperglycemia [6,7]. It is crucial to comprehensively investigate the potential adverse effects of glucose perturbations among PD patients.

Cardiovascular (CV) diseases are the foremost cause of mortality in ESRD patients [8]. Compared to the general population, arterial stiffness is more prevalent in ESRD patients, attributed not only to traditional risk factors but also to non-traditional factors such as uremic toxins, inflammation, endothelial dysfunction, oxidative stress, and mineral bone disease [9,10]. Notably, among ESRD patients undergoing PD, disrupted glucose homeostasis may also play a significant role in the pathogenesis and progression of arterial stiffness. Unfortunately, few previous studies have investigated the association of glucometabolic burdens and aortic stiffness among PD patients [11,12].

The carotid-femoral pulse wave velocity (PWV) is widely recognized as the gold standard in evaluating aortic stiffness [13]. It serves as a robust vascular indicator that independently predicts cardiovascular disease and mortality [14]. This study aimed to explore the relationship between glucometabolic indices, including fasting blood glucose, insulin resistance, advanced glycation end products (AGEs), PD-related glucose exposure, and icodextrin usage, and aortic stiffness in PD patients with and without diabetes mellitus (DM).

## 2. Results

This study included 172 PD patients, of which 110 were non-DM (58 with normal fasting glucose and 52 with impaired fasting glucose) and 62 were DM patients. The mean age of all participants was 58.3 ± 13.5 years, with a median PD duration of 49 months. Six of our participants had previous chronic HD and none of them had a renal transplantation history. The major causes of ESRD were DM nephropathy (36.0%), hypertension (34.9%), and chronic glomerulonephritis (23.3%). The demographic information and clinical characteristics of all participants are summarized in Table 1. Among them, 95 (55.2%) were female, 133 (77.3%) had hypertension, 91 (52.9%) had hyperlipidemia, 5 (2.9%) had atrial fibrillation, and 66 (38.4%) utilized the continuous ambulatory peritoneal dialysis (CAPD) modality. A total of 100 (58.1%) patients were categorized as having high or high average peritoneal equilibration test (PET) scores, and the median total fractional clearance index for urea (Kt/V) was 2.1.

When comparing patients with normal fasting glucose, impaired fasting glucose, and DM, those with normal fasting glucose tended to be younger (*p* < 0.001) and had higher phosphorus levels (*p* = 0.011) and calcium phosphate products (Ca × P) (*p* = 0.039). In contrast, patients with DM had a larger waist circumference (WC) (*p* = 0.017); higher systolic blood pressure (BP) (*p* = 0.005), fasting glucose levels (*p* < 0.001), homeostatic model assessment for insulin resistance (HOMA-IR) scores (*p* = 0.023), and hemoglobin levels (*p* = 0.002); lower total Kt/V (*p* = 0.044); and a higher rate of icodextrin usage (*p* = 0.020).

Regarding aortic PWV among the three groups, DM patients exhibited the highest aortic PWV (9.9 ± 1.9 m/s), followed by those with impaired fasting glucose (9.1 ± 1.4 m/s), while those with normal fasting glucose had the lowest aortic PWV (8.3 ± 1.3 m/s), as illustrated in Figure 1.

Figure 2 illustrates the associations of glucometabolic indices with aortic PWV among non-DM and DM patients. In non-DM patients, individuals in the high HOMA-IR group and high PD glucose load group had significantly higher aortic PWV compared to those in the low HOMA-IR and PD glucose load groups (*p* = 0.007 and 0.010, respectively). No significant differences were observed regarding the high and low serum AGE groups, as well as regarding the use of icodextrin (*p* = 0.776 and 0.615). However, in DM patients, none of the glucometabolic factors mentioned above could be linked to distinct differences in aortic PWV.

Table 2 and Table 3 present the unadjusted and adjusted associations of glucometabolic parameters with aortic PWV in non-DM and DM patients. In non-DM patients, impaired fasting glucose (β = 0.52, 95% confidence interval [CI] = 0.01–1.03, *p* = 0.046), high HOMA-IR scores (β = 0.60, 95% CI = 0.12–1.08, *p* = 0.015), and high PD glucose loads (β = 0.58, 95% CI = 0.08–1.08, *p* = 0.023) were independently associated with higher aortic PWV when compared to the reference groups. In contrast, none of the glucometabolic factors contributed to differences in aortic PWV in DM patients, even in the unadjusted models. The associations stratified by gender are provided in Appendix A.

Subsequently, Figure 3 further provides the aortic PWV values stratified by PD glucose load, fasting glucose status, and HOMA-IR levels among non-DM patients. Patients with impaired fasting glucose and a high PD glucose load exhibited the highest aortic PWV values (*p* for trend < 0.001), as did those with both high HOMA-IR and PD glucose loads (*p* for trend < 0.001). These trends and independent associations are confirmed in Table 4.

## 3. Discussion

In our study, we comprehensively assessed various glucometabolic indices relevant to PD, including fasting glucose, HbA1c, serum AGEs, HOMA-IR, PD glucose load, and icodextrin usage. We sought to investigate the relationships between these indices and aortic stiffness in both DM and non-DM PD patient populations. Our primary finding highlights that, among non-DM PD patients, impaired fasting glucose, insulin resistance, and a high PD glucose load are associated with increased aortic stiffness. These associations remain significant even after rigorous adjustment for potential confounding factors.

Patients with ESRD universally manifest premature vascular aging, which imposes a considerable burden regarding aortic stiffness. Among them, those undergoing PD face an elevated risk of new-onset glucometabolic disorders [15,16,17]. Unsurprisingly, our PD patients with DM exhibited the most pronounced level of aortic stiffness. This observation aligns with the firmly established role of DM in the progression of vascular dysfunction and cardiovascular disease [18,19]. However, it is worth noting that among non-DM patients, those with impaired fasting glucose exhibited a significantly higher aortic PWV compared to those with normal fasting glucose levels. The association between impaired fasting glucose and increased arterial stiffness has been reported in the general population [20,21,22], albeit rarely in the PD population. Importantly, a prior study revealed that non-DM PD patients with impaired fasting glucose had a 2.72-fold higher hazard risk for 2-year mortality when compared to those with normal fasting glucose [23]. These collective findings suggest that, apart from DM status, impaired fasting glucose may hold significant implications for vascular health in PD patients.

Insulin resistance, which increases as renal function decreases in patients with CKD and dialysis [24], is well known to play a dominant role in the pathogenesis of arterial stiffness. Insulin resistance reduces the bioavailability of nitric oxide, promotes low-grade inflammation in the blood vessels and the release of cytokines and chemokines, induces oxidative stress, and causes the overactivation of the renin–angiotensin–aldosterone system, all of which are interconnected and lead to the dysregulated function of vascular smooth muscle, impaired vasodilation, and fibrosis of the vessel wall [25,26]. The association of insulin resistance with either atherosclerosis or arterial stiffness among non-DM patients has been reported in predialysis CKD and ESRD patients undergoing HD [27,28,29]. In line with our results, in non-DM PD patients, insulin resistance has been reported to be associated with aortic stiffness among those aged more than 50 years [11] and to independently predict newly developed CV events and mortality [30,31]. In contrast, HOMA-IR was not associated with aortic stiffness in our DM patients, which implies that, among DM patients undergoing PD, the impact of glucometabolic factors on aortic stiffness is diminished, which could be attributed to the complex and pre-existing heavy burden of vasculopathy in this susceptible group [32,33].

The association between a high PD glucose load and increased aortic stiffness in our non-DM patients indicates the detrimental effects of elevated exposure to PD glucose on vascular health. Among fifty non-DM PD patients, those with coronary calcification were found to have higher total glucose exposure from the dialysis solution [34]. In a study involving one hundred and sixteen PD patients, the glucose load was an independent predictor of endothelial function, assessed by brachial-artery flow-mediated dilation [35]. Increased PD glucose exposure contributes to the enhanced deposition of tissue AGEs, which may adversely affect vascular stiffness and endothelial function [36]. A cross-sectional study in the general population demonstrated that tissue AGEs assessed by skin autofluorescence, but not serum AGEs, were associated with aortic stiffness [37]. Regrettably, the serum AGEs that we measured, rather than tissue AGEs, failed to reveal such an association.

We further explored the association of the PD glucose load and aortic stiffness, stratifying the outcomes based on fasting glucose and the insulin resistance status among non-DM patients. Upon stratification, it was observed that individuals with impaired fasting glucose and high insulin resistance, who simultaneously had high PD glucose exposure, exhibited the highest degree of aortic stiffness. Further investigations are needed to ascertain whether these patients are more vulnerable to vascular injury resulting from a high PD glucose load.

High exposure to peritoneal dialysate glucose has previously been linked to higher all-cause and CV disease mortality in CAPD patients, implying the potential cardiovascular benefits of glucose-sparing dialysate [38,39]. However, the results of our study do not support an association between the use of icodextrin and aortic stiffness. A meta-analysis study has demonstrated that icodextrin is associated with improved ultrafiltration [40]. Regrettably, the low event rates, heterogeneity, and short follow-up duration in the clinical trials included in this analysis do not allow definitive conclusions regarding the use of icodextrin and its potential long-term survival benefits. Further studies are warranted to address this important issue.

This study was subject to several limitations. First, it was a cross-sectional study, which precludes the establishment of causality. Further longitudinal studies are necessary to assess the long-term effects of glucometabolic indices on vascular health. Second, HbA1c or glycated albumin data were unavailable for the non-DM PD group. Third, our DM group was relatively small in size, and the extensive adjustment for all confounding factors may raise concerns about inadequate statistical power. Nevertheless, the lack of an association between glucometabolic indices and aortic stiffness was observed even in unadjusted models, supporting the null association in the DM group. This lack of association may be attributed to the fact that the DM patients had already experienced long-term disruptions in glucose metabolism, resulting in prolonged and heavy damage to their blood vessels due to elevated glucose levels and ultimately progressing to ESRD. Fourthly, all our DM participants had type 2 DM. Therefore, our findings cannot be directly applied to patients with type 1 DM. Finally, there may have been some unmeasured confounding biases affecting the association between the PD glucose load and aortic stiffness, such as daily dietary intake and fluid overload status. Therefore, further longitudinal studies that take these confounding factors into consideration are required to confirm the preliminary findings.

In conclusion, our study highlighted that, among PD patients without DM, aortic stiffness was closely associated with impaired fasting glucose, insulin resistance, and the PD glucose load, which could be regarded as modifiable risk factors for the improvement of CV outcomes. Close monitoring and glucose management by optimizing the PD solution composition and implementing lifestyle modifications, as well as developing innovative therapeutic targets for impaired fasting glucose and insulin resistance, might offer the opportunity to mitigate the progression of aortic stiffness and reduce the adverse effects on vascular health in non-DM patients undergoing PD. On the other hand, the recognition that the DM population experiences worse aortic stiffness emphasizes the necessity for more aggressive management through approaches beyond glucometabolic indices and underscores the importance of developing novel and effective interventions to mitigate the detrimental vascular changes.

## 4. Materials and Methods

### 4.1. Study Design and Population

This cross-sectional study was conducted at Hualien Tzu Chi Medical Center and its three branch hospitals located in Taipei, Taichung, and Dalin. Patient recruitment occurred between February 2020 and May 2021. Participants aged 20 years or older who had been undergoing PD for at least three months were recruited from the nephrology outpatient department. The selection of the dialysis modality was determined by the process of shared decision making based on patient preference. Exclusion criteria comprised acute infections, active malignancies, the presence of a pacemaker or defibrillator, the amputation of limb(s), a bedridden status, or unwillingness to participate. The study received approval from the Institutional Review Board of Tzu Chi Hospital (IRB 108-219-A), and all participants provided written informed consent in compliance with the ethical guidelines outlined in the Declaration of Helsinki before the commencement of the study.

The patient recruitment for the study was unaffected by the COVID-19 outbreak because Taiwan had implemented a ‘zero-COVID’ policy, and outbreaks only began in April 2022. None of the enrolled patients in the study contracted COVID-19.

We collected basic demographic information, including age, gender, and PD duration and modality. Additionally, we recorded the results of the PET and patients’ medical histories, encompassing DM, hypertension, hyperlipidemia, and atrial fibrillation. DM was defined based on a physician’s diagnosis, the use of hypoglycemic medications or insulin, or an HbA1c level ≥ 6.5%. Impaired fasting glucose was characterized as an average fasting blood glucose level within one year ≥ 100 mg/dL among patients without DM.

Furthermore, details regarding medication usage, such as angiotensin-converting enzyme inhibitors (ACEIs) or angiotensin receptor blockers (ARBs), calcium channel blockers (CCBs), calcium carbonate, vitamin D, statins, and icodextrin, were extracted from electronic medical records. The PD glucose load was calculated by determining the total amount of dextrose in the dialysate per day. The PD calorie load was computed based on the total quantity of dextrose (in grams), multiplied by 60% × 3.7 (kcal/g) for the CAPD modality and 40% × 3.7 (kcal/g) for the Automated Peritoneal Dialysis (APD) modality [41].

### 4.2. BP and Aortic PWV Measurements

BP was assessed using an automated upper-arm oscillometric device, with three readings each for systolic BP and diastolic BP taken at the right brachial artery. Aortic PWV values were determined using pressure applanation tonometry with the SphygmoCor system, developed by AtCor Medical (Sydney, NSW, Australia). Participants were instructed to rest in a temperature-controlled environment in a supine position for 15 min before measurements were taken. Aortic PWV measurements were obtained from the carotid-femoral segment, with electrocardiogram signals recorded to establish an R-timing reference. Specialized software was employed to calculate the average time difference between the R-wave and the pulse wave on a beat-to-beat basis, using an average derived from 10 consecutive cardiac cycles [42].

### 4.3. Anthropometric Measurement

Body weight and height were measured for each participant, with an empty abdomen and lightweight clothing. These measurements were taken to the nearest 0.5 kg and 0.5 cm, respectively. Body mass index (BMI) was calculated by dividing the weight in kilograms by the square of the height in meters. WC was measured at the midpoint between the lowest ribs and the iliac crest while the participants stood upright with their hands on their hips.

### 4.4. Biochemical Data

Fasting glucose, hemoglobin, and other routine biochemical data, such as total cholesterol, albumin, total calcium, phosphorus, and intact parathyroid hormone (PTH) concentrations, were obtained from overnight fasting blood samples following standard procedures. The corrected serum calcium levels were determined using the following formula: corrected total calcium (mg/dL) = total calcium (mg/dL) + 0.8 [4 − serum albumin (g/dL)]. Hemoglobin A1C (HbA1c) was available only for individuals with DM. To ensure the robustness of the fasting glucose levels, the average monthly fasting glucose values from the past year were adopted.

Serum AGEs and insulin levels were measured using commercial ELISA kits (Cell Biolabs, San Diego, CA, USA; LDN, Nordhorn, Germany). To assess insulin resistance, we calculated the HOMA-IR score using the following formula: HOMA-IR = (fasting plasma glucose in mg/dL) × (fasting serum insulin in µU/mL)/405 [43]. To determine the Kt/V, we collected 24-h urine and dialysate samples and calculated it according to established protocols [44].

### 4.5. Statistical Analysis

Statistical analyses were conducted using SPSS version 19.0 (SPSS Inc., Chicago, IL, USA). The Kolmogorov–Smirnov test was employed to assess the normal distribution of continuous variables. Variables found to be normally distributed were presented as means ± standard deviation (SD) and analyzed using one-way analysis of variance (ANOVA) among the different groups. Non-normally distributed data were expressed as medians and interquartile ranges, and comparisons were made using the Kruskal–Wallis test. For categorical variables, the Chi-square test was applied. Post-hoc analysis was performed using the Bonferroni test for normally distributed continuous data and categorical data, while Dunn’s test was adopted for non-normally distributed continuous data.

To examine the relationship between glucometabolic variables and cfPWV values in both non-DM and DM patients, univariate and multivariate linear regression analyses were utilized. The covariates adjusted in the multivariate models included age, gender, logarithmically transformed PD duration, hypertension, hyperlipidemia, BMI, WC, systolic BP, calcium phosphate product, total Kt/V, total cholesterol, hemoglobin, albumin, and statin usage. HOMA, serum AGE levels, and PD glucose load, which exhibited a non-normal distribution. Outcomes were categorized into low and high groups based on median values within the non-DM and DM groups for better model fitness and performance. A *p*-value of less than 0.05 was considered significant.

## Figures and Tables

**Figure 1 ijms-24-17094-f001:**
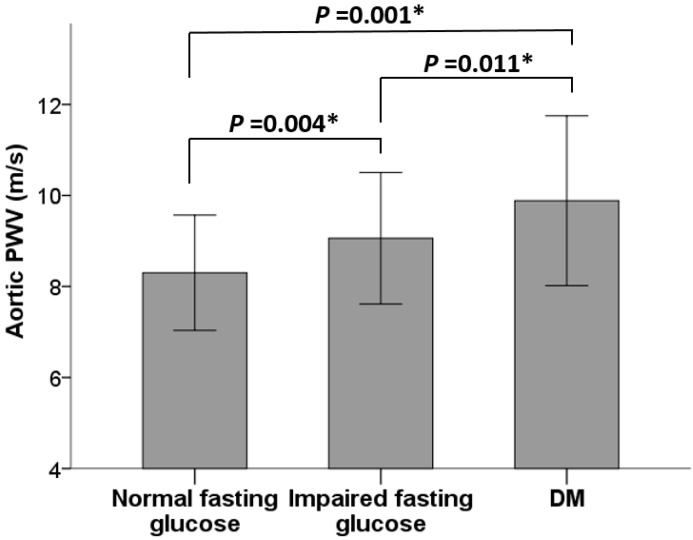
Mean aortic PWV values (±1 SD) among normal fasting glucose, impaired fasting glucose, and DM patients. * *p* < 0.05 was considered to be statistically significant.

**Figure 2 ijms-24-17094-f002:**
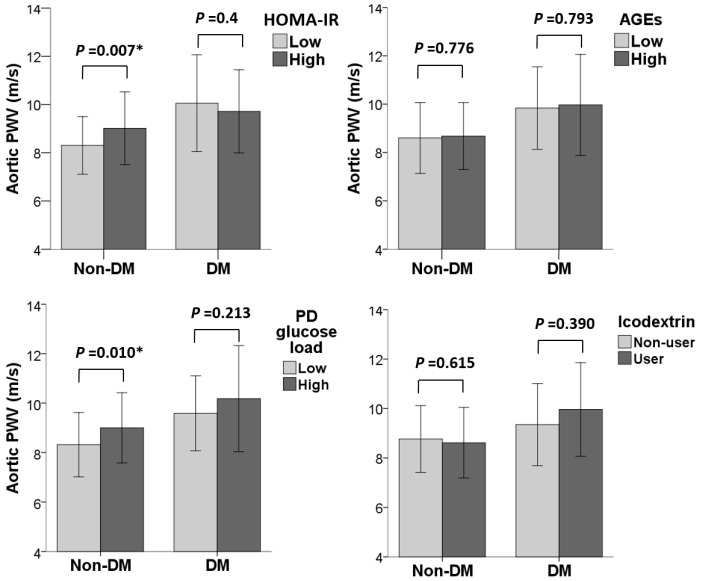
Associations of glucometabolic indices and aortic PWV among non-DM and DM patients. * *p* < 0.05 was considered to be statistically significant.

**Figure 3 ijms-24-17094-f003:**
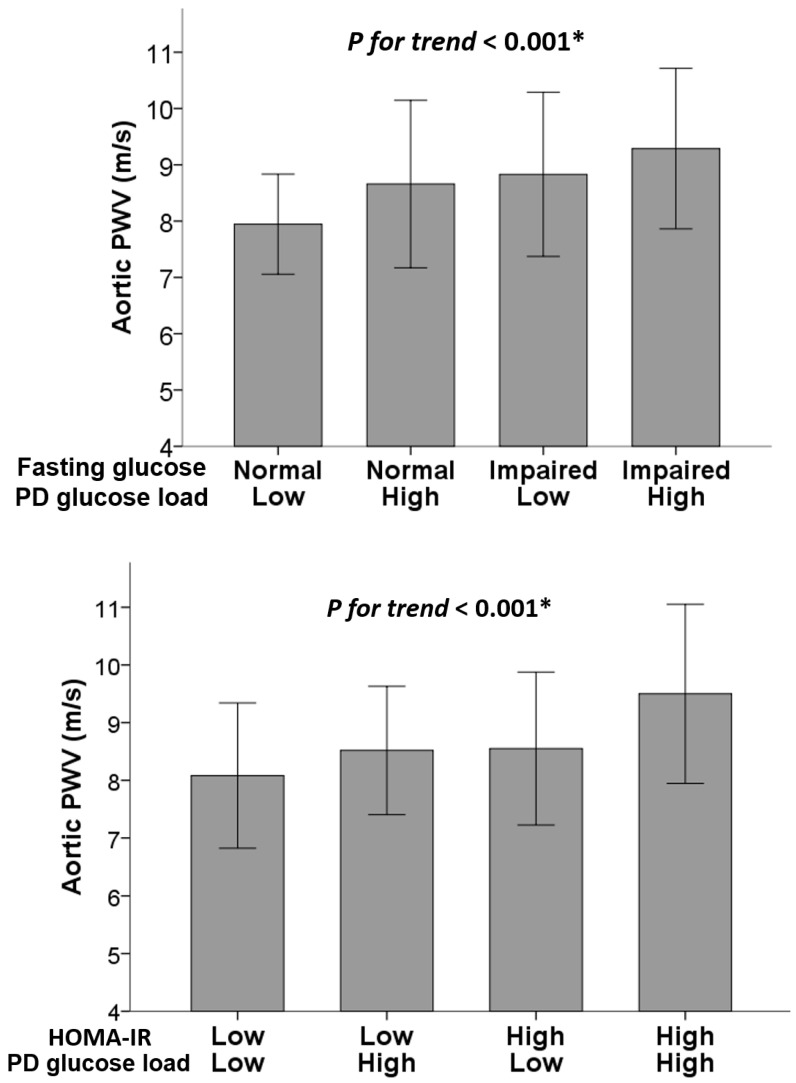
Mean aortic PWV values (±1 SD) stratified based on fasting glucose status, HOMA, and PD glucose load groups among non-DM patients undergoing PD. * *p* < 0.05 was considered to be statistically significant.

**Table 1 ijms-24-17094-t001:** Clinical characteristics of the 172 patients undergoing PD.

Characteristics	All Patients (*n* = 172)	Non-DM		DM (*n* = 62)	*p*
		Normal Fasting Glucose (*n* = 58)	Impaired Fasting Glucose (*n* = 52)		
**Demographics**					
Age (years)	58.3 ± 13.5	51.4 ± 14.7	62.2 ± 12.7 ^a^	61.7 ± 10.1 ^b^	<0.001 *
Female, *n* (%)	95 (55.2)	33 (56.9)	31 (59.6)	31 (50.0)	0.561
PD duration (months)	49 (23–86)	50 (27–78)	57 (27–106)	35 (19–71)	0.063
Hypertension, *n* (%)	133 (77.3)	47 (81.0)	38 (73.1)	48 (77.4)	0.609
Hyperlipidemia, *n* (%) Atrial fibrillation *n* (%)	91 (52.9) 5 (2.9%)	30 (51.7) 1 (1.7%)	26 (50.0) 3 (5.8%)	35 (56.5) 1 (1.6%)	0.770 0.339
**Modality, *n* (%)**					
CAPD	66 (38.4)	20 (34.5)	22 (42.3)	24 (38.7)	0.700
APD or CCPD	106 (61.6)	38 (65.5)	30 (57.7)	38 (61.3)	
**PET status, *n* (%)**					
High or high average	100 (58.1)	34 (58.6)	33 (63.5)	33 (53.2)	0.542
Low or low average	72 (41.9)	24 (41.4)	19 (36.5)	29 (46.8)	
**Examination**					
Weight (kg)	64.4 ± 14.1	63.9 ± 13.4	61.7 ± 13.9	67.1 ± 14.5	0.114
BMI (kg/m^2^)	24.8 ± 3.9	24.5 ± 4.0	24.2 ± 3.9	25.7 ± 3.8	0.080
WC (cm)	92.0 ± 10.4	90.2 ± 9.6	90.5 ± 9.5	95.0 ± 11.2 ^b^	0.017 *
Systolic BP (mmHg)	149 ± 23	144 ± 22	144 ± 24	156 ± 21 ^b,c^	0.005 *
Diastolic BP (mmHg)	85 ± 15	87 ± 16	85 ± 15	83 ± 14	0.360
**Glycemic parameters**					
Fasting glucose (mg/dL)	105 (95–135)	92 (90–96)	107 (104–116) ^a^	145 (128–180) ^b,c^	<0.001 *
Insulin level (ng/mL)	0.67 (0.40–1.03)	0.71 (0.40–1.29)	0.67 (0.51–1.00)	0.65 (0.38–0.93)	0.556
HOMA-IR	5.6 (3.7–8.9)	4.7 (2.6–8.7)	5.1 (4.1–7.6)	6.7 (4.2–10.3) ^b^	0.023 *
AGEs (ng/mL) ^¶^	2.5 (0.7–12.2)	4.2 (0.7–30.0)	2.0 (0.6–7.6)	2.5 (0.7–10.1)	0.219
PD glucose load (g/day)	136 (118–182)	145 (118–182)	136 (103–163)	136 (125–182)	0.105
PD calorie load (Kcal/day)	250 (201–312)	265 (201–323)	228 (176–302)	265 (202–302)	0.318
**Laboratory data**					
Total cholesterol (mg/dL)	172 ± 48	175 ± 44	180 ± 51	162 ± 49	0.121
Hemoglobin (g/dL)	9.8 (8.9–10.6)	9.8 (8.5–10.8)	9.2 (8.4–10.0)	10.2 (9.3–11.0) ^c^	0.002 *
Albumin (g/dL)	3.6 (3.4–3.8)	3.6 (3.4–3.8)	3.6 (3.3–3.9)	3.6 (3.4–3.8)	0.922
Total Kt/V	2.1 (1.8–2.3)	2.1 (1.9–2.3)	2.1 (1.8–2.3)	2.0 (1.8–2.2)	0.044 *
Total Ca, corrected (mg/dL)	9.6 ± 0.8	9.6 ± 0.8	9.6 ± 0.7	9.7 ± 0.7	0.603
Phosphorus (mg/dL)	5.2 ± 1.3	5.6 ± 1.4	5.0 ± 1.3 ^a^	5.0 ± 1.2 ^b^	0.011 *
Ca × P (mg^2^/dL^2^)	50.2 ± 13.4	53.8 ± 14.4	48.2 ± 13.4	48.4 ± 11.8	0.039 *
Intact PTH (pg/mL)	237 (91–485)	269 (117–623)	254 (85–467)	209 (82–350)	0.430
**Medications, *n* (%)**					
ACEIs or ARBs	105 (61.0)	34 (58.6)	28 (53.8)	43 (69.4)	0.215
CCBs	99 (57.6)	30 (51.7)	30 (57.7)	39 (62.9)	0.465
Calcium carbonate	117 (68.0)	42 (72.4)	32 (61.5)	43 (69.4)	0.456
Vitamin D	39 (22.7)	17 (29.3)	13 (25.0)	9 (14.5)	0.137
Statins	52 (30.2)	13 (22.4)	17 (32.7)	22 (35.5)	0.267
Icodextrin	132 (76.7)	38 (65.5)	40 (76.9)	54 (87.1) ^b^	0.020 *

PD, peritoneal dialysis; DM, diabetes mellitus; CAPD, continuous ambulatory peritoneal dialysis; APD, automated peritoneal dialysis; CCPD, continuous cycling peritoneal dialysis; PET, peritoneal equilibrium test; BMI, body mass index; WC, waist circumference; BP, blood pressure; HOMA-IR, homeostatic model assessment for insulin resistance; AGEs, advanced glycation end products; Kt/V, fractional clearance index for urea; Ca, calcium; Ca × P, calcium phosphate product; PTH, parathyroid hormone; ACEIs, angiotensin-converting enzyme inhibitors; ARBs, angiotensin receptor blockers; CCBs, calcium channel blockers. ^a^
*p* < 0.05, comparison between normal and impaired fasting glucose. ^b^
*p* < 0.05, comparison between normal fasting glucose and DM. ^c^
*p* < 0.05, comparison between impaired fasting glucose and DM. * *p* < 0.05 was considered to be statistically significant among three groups. ^¶^ Serum AGE levels were not measured in six patients.

**Table 2 ijms-24-17094-t002:** Associations of glycemic parameters with aortic PWV among non-DM patients undergoing PD.

Variables	Aortic PWV (m/s)			
	Unadjusted		Adjusted	
	β (95% CI)	*p*	β (95% CI)	*p*
**Fasting glucose**				
Normal (<100 mg/dL)	Reference	0.004 *	Reference	0.046 *
Impaired (≥100 mg/dL)	0.76 (0.25, 1.27)		0.52 (0.01, 1.03)	
**HOMA-IR status**				
Low (≤4.91)	Reference	0.007 *	Reference	0.015 *
High (>4.91)	0.71 (0.20, 1.23)		0.60 (0.12, 1.08)	
**Serum AGE level ^¶^**				
Low (≤2.5 ng/mL)	Reference	0.667	Reference	0.777
High (>2.5 ng/mL)	0.12 (−0.42, 0.65)		0.07 (−0.43, 0.57)	
**PD glucose load**				
Low (≤136 g/day)	Reference	0.010 *	Reference	0.023 *
High (>136 g/day)	0.68 (0.16, 1.20)		0.58 (0.08, 1.08)	
**Icodextrin**				
Non-user	Reference	0.615	Reference	0.739
User	−0.15 (−0.73, 0.44)		−0.09 (−0.63, 0.45)	

The adjusted models adopted age, gender, log-PD duration, hypertension, hyperlipidemia, BMI, WC, systolic BP, Ca × P, total Kt/V, total cholesterol, Hb, albumin, and statins as covariates. We classified our groups as low and high based on the median values within non-DM patients. ^¶^ Serum AGE levels were not measured in four patients. PWV, pulse wave velocity; DM, diabetes mellitus; PD, peritoneal dialysis; CI, confidence interval; HOMA-IR, homeostatic model assessment for insulin resistance; AGEs, advanced glycation end products. * *p* < 0.05 was considered to be statistically significant.

**Table 3 ijms-24-17094-t003:** Associations of glycemic parameters with aortic PWV among DM patients undergoing PD.

Variables	Aortic PWV (m/s)			
	Unadjusted		Adjusted	
	β (95% CI)	*p*	β (95% CI)	*p*
**HbA1c (%)**				
≤7%	Reference	0.161	Reference	0.362
>7%	−0.69 (−1.67, 0.29)		−0.48 (−1.52, 0.57)	
**HOMA-IR status**				
Low (≤6.66)	Reference	0.483	Reference	0.860
High (>6.66)	−0.34 (−1.29, 0.62)		0.09 (−0.92, 1.09)	
**Serum AGE level ^¶^**				
Low (≤2.5 ng/mL)	Reference	0.849	Reference	0.628
High (>2.5 ng/mL)	0.09 (−0.86, 1.05)		−0.24 (−1.24, 0.76)	
**PD glucose load**				
Low (≤136 g/day)	Reference	0.213	Reference	0.551
High (>136 g/day)	0.59 (−0.35, 1.54)		0.36 (−0.85, 1.57)	
**Icodextrin**				
Non-user	Reference	0.390	Reference	0.667
User	0.61 (−0.80, 2.03)		0.32 (−1.18, 1.83)	

The adjusted models adopted age, gender, log-PD duration, hypertension, hyperlipidemia, BMI, WC, systolic BP, Ca × P, total Kt/V, total cholesterol, Hb, albumin, and statins as covariates. We classified our groups as low and high based on the median values within DM patients. ^¶^ Serum AGE levels were not measured in two patients. PWV, pulse wave velocity; DM, diabetes mellitus; PD, peritoneal dialysis; CI, confidence interval; HOMA-IR, homeostatic model assessment for insulin resistance; AGEs, advanced glycation end products.

**Table 4 ijms-24-17094-t004:** Exploratory analysis of interaction effects of fasting glucose, insulin resistance, and PD glucose load on aortic PWV among non-DM patients undergoing PD.

**Group**		**N**	**Aortic PWV (m/s)**			
**Fasting Glucose**	**PD Glucose Load**		**Unadjusted**		**Adjusted**	
			**β (95% CI)**	***p* for trend**	**β (95% CI)**	***p* for trend**
Normal	Low	29	Reference	<0.001 *	Reference	0.012 *
Normal	High	29	0.71 (0.02, 1.41)		0.72 (0.07, 1.37)	
Impaired	Low	26	0.89 (0.17, 1.60)		0.79 (0.12, 1.45)	
Impaired	High	26	1.34 (0.63, 2.06)		0.84 (0.16, 1.51)	
**HOMA-IR**	**PD glucose load**	**N**	**Unadjusted**		**Adjusted**	
			**β (95% CI)**	***p* for trend**	**β (95% CI)**	***p* for trend**
Low	Low	27	Reference	<0.001 *	Reference	0.001 *
Low	High	28	0.44 (−0.27, 1.14)		0.51 (−0.18, 1.19)	
High	Low	28	0.47 (−0.24, 1.18)		0.52 (−0.14, 1.19)	
High	High	27	1.42 (0.71, 2.13)		1.33 (0.62, 2.05)	

The adjusted models adopted age, gender, log-PD duration, hypertension, hyperlipidemia, BMI, WC, systolic BP, Ca × P, total Kt/V, total cholesterol, Hb, albumin, and statins as covariates. We classified our groups as low and high based on the median values within non-DM patients. PD, peritoneal dialysis; PWV, pulse wave velocity; DM, diabetes mellitus; CI, confidence interval; HOMA-IR, homeostatic model assessment for insulin resistance. * *p* < 0.05 was considered to be statistically significant.

## Data Availability

The data presented in this study are available on request from the corresponding author.

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
