# Peer review of "Associations of Glucometabolic Indices with Aortic Stiffness in Patients Undergoing Peritoneal Dialysis with and without Diabetes Mellitus"

_ijms, 2023, doi:10.3390/ijms242317094_

Round 1

Reviewer 1 Report

Comments and Suggestions for Authors

The authors present a nice application of glucometabolic indices in peritoneal dialysis and their correlation to aortic stiffness.

There are some points which need to be resolved:

References of the sentence line 57-58 are misssing (Unfortunately, few previous studies...among PD patients.)

The classification of HOMA-IR status, serum AGEs level and PD glucose load in "low" and "high" was based on median values of the total population or subgroups?

Table 1, Clinical characteristics is presenting mean values or median values especially regarding glycemic parameters?

Patients with atrial fibrillation had been included ?

Reviewer 2 Report

Comments and Suggestions for Authors

The subject is clinically important, the study is well designed, the results are presented in a clear way, the discussion contains the desirable level of self-criticism.

The clinical data concerning PD should be supplemented. This method is definitely underscored among adults with CKD. Thus, when the vast majority of ESRD patients undergo hemodialysis, it is worth underlining the indications for PD.

In detail, for how many of these patients PD was their first method of renal replacement therapy? How many of them have already had kidney transplantation/ previous chronic hemodialysis ? What was the underlying disease leading to CKD?

Reviewer 3 Report

Comments and Suggestions for Authors

The manuscript is well written. However, the authors are requested to address the following queries:

1.      What is novelty of the study?

2.      Do the authors have data regarding male vs. female and type 1 vs type 2 diabetic patients?

3.      “In conclusion, among PD patients without DM, impaired fasting glucose, insulin resistance, and PD glucose load were closely associated with aortic stiffness. In contrast, none of the glucometabolic factors contributed to differences in cfPWV in DM patients.” How did the authors confirm that glucometabolic factors didn’t contribute to differences in cfPWV in DM patients and what could be the potential reason of this observation?

4.     “Closely monitoring and glucose management by optimizing the PD solution composition, implementing lifestyle modifications, as well as developing innovative therapeutic targets for impaired fasting glucose and insulin resistance might have the opportunity to mitigate the progression of aortic stiffness and reduce adverse effects on vascular health in non-DM patients undergoing PD”. How will the DM patients undergoing PD be benefitted from this particular study?

5.     The study was performed on the patients during 2020-21, the peak of the ‘Covid-19’pandemic. How could Covid-19 have influenced on the outcome of the study? Were the authors aware about this important aspect? Please discuss.

6.     How long the patients went through the PD?

Comments on the Quality of English Language

The manuscript is well written. However, the authors are requested to address the following queries:

1.      What is novelty of the study?

2.      Do the authors have data regarding male vs. female and type 1 vs type 2 diabetic patients?

3.      “In conclusion, among PD patients without DM, impaired fasting glucose, insulin resistance, and PD glucose load were closely associated with aortic stiffness. In contrast, none of the glucometabolic factors contributed to differences in cfPWV in DM patients.” How did the authors confirm that glucometabolic factors didn’t contribute to differences in cfPWV in DM patients and what could be the potential reason of this observation?

4.     “Closely monitoring and glucose management by optimizing the PD solution composition, implementing lifestyle modifications, as well as developing innovative therapeutic targets for impaired fasting glucose and insulin resistance might have the opportunity to mitigate the progression of aortic stiffness and reduce adverse effects on vascular health in non-DM patients undergoing PD”. How will the DM patients undergoing PD be benefitted from this particular study?

5.     The study was performed on the patients during 2020-21, the peak of the ‘Covid-19’pandemic. How could Covid-19 have influenced on the outcome of the study? Were the authors aware about this important aspect? Please discuss.

6.     How long the patients went through the PD?

Round 2

Reviewer 3 Report

Comments and Suggestions for Authors

The manuscript has been revised thoroughly and improved a lot.